# Wastewater Surveillance in Europe for Non-Polio Enteroviruses and Beyond

**DOI:** 10.3390/microorganisms11102496

**Published:** 2023-10-05

**Authors:** Laura Bubba, Kimberley S. M. Benschop, Soile Blomqvist, Erwin Duizer, Javier Martin, Alexander G. Shaw, Jean-Luc Bailly, Lasse D. Rasmussen, Anda Baicus, Thea K. Fischer, Heli Harvala

**Affiliations:** 1European Non-Polio Enterovirus Network (E.N.P.E.N.), 1207 Geneva, Switzerland; 2National Institute for Public Health and the Environment (RIVM), 3721 MA Bilthoven, The Netherlands; kim.benschop@rivm.nl (K.S.M.B.); erwin.duizer@rivm.nl (E.D.); 3Finnish Institute for Health and Welfare, P.O. Box 95, 70701 Kuopio, Finland; soile.blomqvist@thl.fi; 4Division of Vaccines, Medicines and Healthcare Products Regulatory Agency, Potters Bar EN6 3QG, UK; javier.martin@mhra.gov.uk; 5MRC Centre for Global Infectious Disease Analysis, London SW7 2AZ, UK; a.shaw@imperial.ac.uk; 6Abdul Latif Jameel Institute for Disease and Emergency Analytics, School of Public Health, Imperial College London, London SW7 2BX, UK; 7Laboratoire Micro-Organismes Genome Environnement (LMGE), Université Clermont Auvergne CNRS, 63001 Clermont-Ferrand, France; j-luc.bailly@uca.fr; 8Virus Surveillance and Research Section Department of Virus and Microbiological Special Diagnostics Statens Serum Institut, DK-2300 Copenhagen, Denmark; lara@ssi.dk; 9Enteric Viral Infections Laboratory, Cantacuzino National Institute for Medical-Military Research and Development, 020123 Bucharest, Romania; baicus.anda@cantacuzino.ro; 10Department of Clinical Research, University Hospital of Nordsjaelland, 3400 Hilleroed, Denmark; 11Department of Public Health, University of Copenhagen, 1172 Copenhagen, Denmark; 12Microbiology Services National Health Service (NHS) Blood and Transplant, London NW9 5BG, UK; heli.harvalasimmonds@nhsbt.nhs.uk; 13Division of Infection and Immunity, University College London, London WC1E 6BT, UK

**Keywords:** wastewater surveillance, environmental surveillance, non-polio enterovirus, poliovirus

## Abstract

Wastewater surveillance (WWS) was developed in the early 1960s for the detection of poliovirus (PV) circulation in the population. It has been used to monitor several pathogens, including non-polio enteroviruses (NPEVs), which are increasingly recognised as causes of morbidity in children. However, when applying WWS to a new pathogen, it is important to consider the purpose of such a study as well as the suitability of the chosen methodology. With this purpose, the European Non-Polio Enterovirus Network (ENPEN) organised an expert webinar to discuss its history, methods, and applications; its evolution from a culture-based method to molecular detection; and future implementation of next generation sequencing (NGS). The first simulation experiments with PV calculated that a 400 mL sewage sample is sufficient for the detection of viral particles if 1:10,000 people excrete poliovirus in a population of 700,000 people. If the method is applied correctly, several NPEV types are detected. Despite culture-based methods remaining the gold standard for WWS, direct methods followed by molecular-based and sequence-based assays have been developed, not only for enterovirus but for several pathogens. Along with case-based sentinel and/or syndromic surveillance, WWS for NPEV and other pathogens represents an inexpensive, flexible, anonymised, reliable, population-based tool for monitoring outbreaks and the (re)emergence of these virus types/strains within the general population.

## 1. Introduction

Enteroviruses (EVs) are members of the Picornaviridae family, with over 110 types known to infect humans, belonging to species A–D. The most well-known EV is poliovirus (PV), belonging to species C, which can cause acute flaccid paralysis (AFP, also known as polio or poliomyelitis). PV encompasses three serotypes, and thanks to the effort of the Global Polio Eradication Initiative (GPEI) programme, wild-type PV (WPV) is currently only endemic in Afghanistan and Pakistan [1]. However, occasional vaccine-derived PV (VDPV) detections are reported worldwide, including in Europe [2,3,4,5], underlining the necessity to maintain a strong population immunity and a strong disease surveillance, as stated by the World Health Organisation (WHO) [6].

Wastewater surveillance has been improved over the years, and one of useful criterion for satisfactory overall performance is the detection of non-polio enteroviruses (NPEVs). In fact, at least 30% of concentrated sewage from grab samples and at least 10% of traps should reveal non-polio enterovirus [7]. Thus NPEV should be always be detected in wastewater and can be used as control of the overall procedures. Alternatively, it has been suggested that pepper mild mottle virus, present in human faeces, could be used as a control [8].

Though NPEVs may cause asymptomatic infections, they have also been increasingly recognised as the cause of severe diseases. An example is EV-D68, which has been recognized as cause of polio-like paralytic disease (now known as acute flaccid myelitis, AFM) [9,10,11]. NPEVs have also been associated with other severe diseases, such as aseptic meningitis, encephalitis, hand, foot, and mouth disease (HFMD), and myocarditis, mostly affecting children under 5 years of age [12,13,14,15]. Contrary to PV, information on NPEV’s morbidity and mortality burden is limited due to a lack of systematic implementation of clinical data, virological diagnosis, and typing within a standardized surveillance system.

The European Non-Polio Enterovirus Network (ENPEN) [16] was established to study the disease burden of NPEVs under the auspices of the European Society for Clinical Virology (ESCV) in 2018 and has brought together over 70 hospitals and public health institutes from 20 European countries. ENPEN aims to raise professional awareness for more effective detection and typing, to develop diagnostics, and to guide surveillance activities. Sharing data within the network has helped to detect outbreaks and emerging enterovirus types, enhancing public health response and guiding future evidence-based treatment and prevention strategies for adequate clinical management of enterovirus infections [16]. To date, ENPEN has contributed to evaluate, define and describe the circulation of EVs and parechovirus, to assess the methodologies, to recommend nomenclature, and to help in implementing surveillance systems [17,18,19,20,21,22].

Systematic clinical surveillance and wastewater surveillance for NPEV are limited or absent across the European area, despite formal wastewater surveillance guidance for PV having been published in 1990 [23]. Wastewater surveillance is defined as anonymised screening of large groups through what is excreted via stool or urine. It is a powerful tool to rapidly detect the circulation of pathogens, particularly if the pathogens in question circulate largely asymptomatically and are therefore unlikely to be noted in case-based testing or via clinical surveillance. Whilst there is currently no systemic wastewater surveillance for NPEVs, it should be noted that numerous data on NPEVs’ circulation could be generated from PV data [24,25,26]. Whether utilising existing wastewater surveillance systems or establishing a dedicated network for targets such as NPEVs, there are many technical aspects to consider, for example, how to best sample sewage, what methods to use for concentration of collected wastewater, and which assays can be used to maximize sensitivity and range of NPEV types in the detection process. In addition, SARS-CoV-2 circulation and the necessity to tackle the pandemic revived the use of wastewater surveillance, multiplying the methodologies in use, sometimes without proper validation of the methodology.

For these reasons, ENPEN organised an on-line symposium to discuss the optimal systems for wastewater surveillance, including sampling and virus identification methods. Here, we report the main findings, exploring the history and the challenges of wastewater surveillance initially introduced for PV detection in Europe and highlighting its applications and limitations for the surveillance of NPEVs. We also discuss the optimal use of wastewater surveillance as a supplement to existing surveillance systems and its relevance for the identification of other pathogens.

## 2. Materials and Methods

### Implementation of Wastewater Surveillance Methodology in Finland

Wastewater surveillance, traditionally used to monitor the circulation of PV, was implemented in Finland in the early 1960s, when the national inactivated poliovirus vaccine (IPV) programme was launched. At that time, despite the absence of clinical cases of AFP, 6% of the sewage samples tested positive for PV; the disappearance of PV from sewage was only noted in the late 1960s or early 1970s [27]. In 1982, PV wastewater surveillance was discontinued, to be reintroduced only two years later when an outbreak of wild poliovirus type 3 (WPV3) of Mediterranean origin was observed in Finland. This involved a total of 10 confirmed poliomyelitis cases, and an estimation of over 100,000 people infected [22]. This outbreak not only highlighted the importance of continuing wastewater surveillance during periods when no polio cases were identified for faster control of outbreak, but also provided an opportunity, for the first time, to validate the wastewater surveillance methodology, both experimentally and mathematically.

With this purpose, a 10^7−5^ CCID 50/mL of vaccine strain of PV-1 (Sabin) cultivated on HEp2 was introduced into the Helsinki sewage network via a single toilet flush. Sewage samples were collected at a downstream location in subsequent days for further virus analysis [28]. This experiment was repeated three times, at one-month intervals, demonstrating the detection of the virus for four days after the single flush. It was calculated that 400 mL of sewage sample was sufficient for the detection of virus if one out of 10,000 inhabitants was excreting the virus in a population of 700,000 people. The concentration trend changed over time, with the first two days showing a higher concentration of virus particles. A further mathematical model was developed to predict how the mode of transmission, environmental factors, and laboratory methods would affect the recovery of PV, demonstrating that bi-weekly analysis of one sewage sample in a population of 700,000 would allow the detection of an emerging PV within a few months [29].

Intensive wastewater surveillance performed during the Finnish 1984–1985 outbreak revealed a wide geographical spread of PV, even in the regions where clinical cases were not seen. It demonstrated that this kind of surveillance can be used as a routine tool to monitor virus circulation.

An oral PV vaccine was distributed to all Finnish citizens between February and March 1985, as a public health measure aiming to limit the spread of polio. The subsequent wastewater surveillance highlighted the presence of OPV strains in the sewage, with the last detection dated September 1985 [30]. Finland has continued wastewater surveillance since, and in 2015, the WHO recommended wider implementation of wastewater surveillance to support the end-game of polio eradication [31].

## 3. Results

### 3.1. Application of Wastewater Surveillance

In addition to PV, wastewater surveillance has been used to screen for other pathogens, such as NPEVs, SARS-CoV-2, measles, and gastrointestinal viruses such as norovirus, adenovirus, and rotavirus [24,32,33,34,35]. Before applying wastewater surveillance for any pathogen, it is important to consider that the likelihood of detection is enhanced if the pathogen is shed in high viral loads either in faeces or urine, allowing an easier estimation of the scale of viral circulation in cases where asymptomatic infections form the highest proportion of all infections. In addition, mitigation measures should be available to limit pathogen circulation if identified via wastewater surveillance. Wastewater surveillance can be applied for two main purposes: firstly, to enable early detection of an introduction or emergence of pathogens (absence or presence) and to monitor their spread; secondly, to monitor the changes in transmission and prevalence of highly prevalent/endemic pathogens by using quantifiable methods in testing. For the first purpose (see examples in “Targeted Wastewater Surveillance for Early Detection of Viruses”), a sensitive method able to target a small population and detect a single shedder should be implemented, as was demonstrated with PV. For the second purpose (see example in “General Wastewater Surveillance”), surveillance should aim to represent the whole population and should consider the use of a simple and fast method for the pathogen detection, as it was done for SARS-CoV-2 [35]. In addition, wastewater surveillance can be used to detect any breaches of containment (see examples in “Wastewater Surveillance to Enhance Laboratory Safety”).

#### 3.1.1. Examples of Wastewater Applications

Targeted Wastewater Surveillance for Early Detection of Viruses

Examples of targeted approaches include surveillance of evacuees and refugee camps in the Netherlands, where a high number of people originate from PV-endemic countries, like Afghanistan. In this case, sample collection is often driven by PV outbreaks in the originating country, and it is used to target other endemic viruses in the region, such as HAV and measles.

An additional example in the Netherlands is wastewater surveillance targeting villages and schools in the so-called Bible Belt, a region where PV vaccination coverage is low due to religious reasons. Grab sampling is conducted every 6 weeks, and samples are screened for EV, gastrointestinal viruses, coronavirus, and measles when there are clinical cases of the latter [36]. The aim is to have an early signal of polio or other virus introduction into this immune-sensitive population and provide an estimation of the scale of circulation. This is similar for measles viruses, where detection and circulation are often related to a single point of introduction [24].

Another example is from the United States, where wastewater surveillance is enhanced when cases of polio or poliovirus are detected. In 2022, a case presenting with paralysis tested positive for vaccine-derived poliovirus (VDPV2) in New York [37]. As a result, intensified wastewater surveillance demonstrated that the virus was actually present in sewage prior to the identification of clinical cases (as shown in Finland previously). Similarly, in Romania, PV and NPEV wastewater surveillance was implemented in 2015, following the possible introduction of VDPV1 and VDPV2 from Ukraine. For that purpose, wastewater collection and a commercial real-time RT-PCR method were implemented. A total of 10 mL of sewage was concentrated by centrifugation at 1500 g in a refrigerated centrifuge for 10 minutes [38].

General Wastewater Surveillance

For endemic viruses, such as non-polio enteroviruses and other gastrointestinal viruses, wastewater surveillance is used routinely to complement national clinical surveillance, not as an early warning system. As a result of the COVID-19 pandemic, environmental surveillance was implemented and has expanded to sample all sewage treatment plants in the Netherlands. Sampling is conducted 3 to 4 times a week and includes 24-hour composite samples rather than a grab raw-sewage sampling. Analysis is based on nucleic acid extracted from raw wastewater, and the results are used to estimate the incidence of SARS-CoV-2 in the general population. Due to the lack of a labour-intensive concentation step, the method is less sensitive, but it is faster and simpler and is geared to provide incidence estimates rather than early detection.

Wastewater Surveillance to Enhance Laboratory Safety

In the Netherlands, environmental surveillance is also applied at Poliovirus Essential Facilities where infectious poliovirus is handled. The aim of this surveillance is to detect any breaches of containment. In April 2016, the Global Action Plan (GAP) III/IV came in effect with the mandate to contain post-eradication polioviruses under strict conditions. Currently, PV type 2 (Sabin and wild) and WPV3 are declared eradicated and fall under GAPIII/IV containment requirements.

Environmental surveillance at the facilities with such viruses should include sampling of wastewater downstream of facilities every 3 weeks, and in case an infectious virus is detected, an immediate response by the facility and public health institutes is required [39].

### 3.2. Application of Wastewater Surveillance Procedure and Ethods

Wastewater surveillance was initially launched as a complementary tool for PV eradication, with different protocols established in endemic and polio-free regions. Compared to testing on clinical samples, the analysis of wastewater poses more challenges due to the low viral amount that may be recovered and the matrix effects on detection as a consequence of the complex components present in wastewater. Thus, the purification and concentration of wastewater is essential to obtain a viral concentration sufficient to be detected and analysed. The general procedure consists of the collection of a high volume of grabs, generally around 500 mL, which are then clarified with a two-phase separation method using polymers and overnight separation, followed by chloroform extraction and virus isolation in cell culture (Figure 1) [7]. This method, firstly applied and adjusted during the 1984–1985 outbreak in Finland, allows obtaining a 50–100-fold concentrate with an interphase where the EV accumulates. Furthermore, a comparison of pathogen detection by PCR after RNA extraction between raw and concentrated sewage showed a higher detection rate of enteroviruses, including poliovirus, and other enteric viruses in the concentrated samples, even though a higher degree of inhibition was also observed [39].

Alternatively, precipitation with polyethylene glycol (PEG) with a day-long centrifuge and ultra-filtration have been used to obtain more than a 100-fold nominal concentration [39].

More recently, due to the SARS-CoV-2 pandemic, several additional methods have been proposed and performed [40] to concentrate wastewater, with the aim to reduce the timing dedicated to the procedure, e.g., ultracentrifugation [41]. This latter method consists of centrifugation at 200,000× *g* for 1 h at 4 °C for 11 mL of wastewater.

Molecular methods, such as real time RT-PCR and sequencing approaches, can be applied after RNA extraction for detection and characterisation of PV, NPEV, and other pathogens [24,32,33,34,35].

### 3.3. Improving Wastewater Surveillance—Direct Detection by NGS

Whilst poliovirus wastewater surveillance still typically employs virus isolation as the gold standard method, direct (cultureless) molecular detection methods offer some distinct advantages. Firstly, avoiding amplification of the virus in culture is in line with stricter containment of PV, as the world moves towards eradication. Secondly, methods that can provide an amplicon and undergo next-generation sequencing (NGS) allow the determination of multiple virus variants, including different serotypes identified from the same sample, as there is no competitive growth nor the requirement for a single template for sequencing. Direct molecular detection with a long-read NGS method (such as Nanopore sequencing) has demonstrated its utility for environmental poliovirus surveillance, allowing the identification of closely related virus variants from one sewage sample [42].

Molecular methods, however, may struggle to match the sampling volume possible using cell culture. For example, for environmental poliovirus surveillance, eight tissue culture flasks each test 0.5 mL of sewage concentrate, as opposed to a typical 5–8 µL applied for an RT-PCR reaction. Methods to achieve greater concentration of viruses together with better nucleic acid extraction to reduce the high level of PCR inhibition will be important to allow molecular methods to survey a larger volume of the sewage sample, hence increasing their sensitivity. In contrast, the use of molecular methods for the direct detection of NPEVs from sewage samples is advantageous when only a small proportion of NPEV strains can be isolated in cell cultures typically used in clinical diagnostic laboratories. Furthermore, molecular methods are also needed to distinguish a vaccine-like PV virus and circulating VDPV from each other.

Direct detection (i.e., culture-free) methods based on nucleic acid extraction and PCR followed by Sanger sequencing have been available for a number of years, but the use of more recent molecular methods in combination with NGS analysis directly from environmental samples has started to unravel the complexity of the virome in wastewater samples. Direct detection of NPEVs has been demonstrated through genus-wide and serotype specific RT-PCR, with NGS again allowing resolution of the vast array of viruses that may be present when using broadly targeted primers. A few years ago, a novel method was developed to allow direct sequencing of the entire capsid coding region of EVs from all four enterovirus species, A to D, and successfully identify several strains from multiple serotypes present in a single sewage sample [43]. This method was later adapted to a nanopore long-read sequencing [44,45].

General considerations on the timing and limitations of the methods described are reported in Table 1.

### 3.4. Wastewater Surveillance during SARS-CoV-2 Pandemic

The ongoing SARS-CoV-2 pandemic demonstrates the ease of adapting lessons learned from previous PV wastewater surveillance for other pathogens.

When SARS-CoV-2 started to circulate worldwide, many countries, including Australia, Finland, France, India, Israel, the Netherlands, Sweden, Ukraine, and Denmark, implemented the monitoring of untreated wastewater from urban areas as a supplementary surveillance tool for large-scale assessment of SARS-CoV-2 spread in the community. At the beginning of the COVID-19 pandemic, the initial objective was the early detection of the presence of the coronavirus in catchment populations selected throughout a country to provide data on virus circulation to policy makers. As the pandemic progressed and the public health risks changed, this objective evolved into the monitoring of trends of virus circulation in the general population and the identification of SARS-CoV-2 variants.

While several countries already had wastewater surveillance in place for poliovirus surveillance and could hence easily adapt that platform for SARS-CoV-2, other countries, like Denmark, where no wastewater surveillance of disease-causing agents of any kind existed, had to implement the system from the beginning. In Denmark, prior to the implementation of wastewater surveillance, legislation had to be changed in order to grant health authorities permission to collect and analyse wastewater. A pilot project was initiated covering the seven wastewater treatment plants on the island of Bornholm, where restrictions were eased earlier than in the rest of the country. Samples were taken daily for a 30-day period. This was in a period with very low human incidences, but nonetheless, the detection of SARS-CoV-2 was possible in over 10% of the samples, especially in samples collected late in the project period, during Easter, where many tourists visited the island. A second pilot study was initiated when the Alpha variant was being replaced by the Delta variant. Pilot 2 included 20 sampling sites covering all of Denmark, with varying catchment areas. Sequencing of a 1049 bp amplicon of the spike gene showed that detection and differentiation of variants in environmental samples containing a mixture of variants was possible. Immediately after Pilot 2, full-scale surveillance was initiated, where 230 sampling sites with catchment area populations between 700and 1 million, covering 85% of the Danish population, were sampled three times a week, and the samples with the highest viral load from each sampling site were sequenced. Results were published in a weekly report [46].

Several important lessons can already be drawn in terms of benefits, limitations, and improvements from the experiences of implementation of SARS-CoV-2 surveillance from these countries. Wastewater testing for quantitative detection of SARS-CoV-2 RNA provides near real-time, granular, and evidence-based data not influenced by screening strategies on virus circulation within catchment populations, whether these trends are caused by changes in SARS-CoV-2 variants, seasonal tourism movements, or preventive and active intervention measures implemented against COVID-19. By calculating metrics from viral RNA concentrations in sequential wastewater samples, it is also possible to assess variations in virus circulation, allowing prediction of a possible increase in the demand of health care in the coming days or weeks [31]. However, SARS-CoV-2 infection presents unique challenges, limiting the application of wastewater-based epidemiology to COVID-19. For instance, the infection ratio may be different for each variant and could be vaccination rate–dependent. Also, timing and duration of viral shedding is variable among symptomatic and asymptomatic individuals. Thus, it is not feasible to provide reliable assessment of the number of COVID-19 cases in the general population from wastewater quantitative data. Although, these limitations should be overcome, monitoring the presence of SARS-CoV-2 in wastewater can be considered as an operational tool for the management of epidemic response, which meets public health objectives.

## 4. Discussion

Surveillance through wastewater monitoring and testing was introduced decades ago and has recently proven its worth for other diseases, including during the SARS-CoV-2 pandemic, where it replaced expensive and resource-extensive individual testing in several countries.

In addition, in 2022, it was proven that the establishment of wastewater surveillance was essential to identify the circulation of VDPV2 in New York [37] and London [4], allowing the implementation of public health actions that included enhancement of vaccination. However, it needs to be noted that the situation in the two metropolises was very different. In New York, the alert started from a clinical case presenting paralytic poliomyelitis in an unvaccinated adult, and the sequential investigation of the wastewater highlighted the circulation of PV in the sewage. Later, typing confirmed in some cases the genetic link with the virus infecting the clinical case [37]. On the contrary, the routine surveillance in the UK allowed early detection of the circulation of VDPV2 before the identification of any clinical cases. The early identification allowed the implementation of an enhanced surveillance and an inactivated polio vaccine campaign among children. Also, the whole genome sequencing analysis established a genetic link between the isolates and a recombinant PV lineage, which was the same detected in New York [47].

Wastewater surveillance is not intended to replace clinical disease surveillance based on individual cases and contact tracing, but it is a fairly inexpensive supplemental data source that can support traditional epidemiological surveillance. It can also help overcome the well-known limitations of clinical surveillance, including low population coverage, limited testing availability, and reporting delays, as well as challenges posed by asymptomatic carriage.

With poliovirus eradication getting closer, the current surveillance based on individual detection of severe cases of polio will become a suboptimal surveillance strategy. However, sustaining a wastewater surveillance system for just one pathogen that is supposed to be absent will not be cost-effective. Combining efforts to use modern molecular-based wastewater surveillance for multiple pathogens to complement clinical surveillance will make the systems more cost-efficient and sustainable.

Wastewater surveillance, however, may also be the starting point for the early identification of zoonotic pathogens that can potentially spill over into the human population. In fact, nowadays, the pathogen research in this matrix has been limited to those pathogens already circulating in the human population, but the implementation of wastewater surveillance in farm areas, accounting for the majority of zoonosis with pathogens infecting multiple hosts, may allow the early identification of public health threats [48].

This approach would be in line with the increasing recognition that a health framework could be the most effective tool for the early detection of pathogens potentially causing an epidemic or even a pandemic. Therefore, it appears relevant to improve shared policies and protocols, although laboratory capacity and resources can limit the implementation of standardized methods for the detection of pathogens through wastewater surveillance. This would allow perpetrating a more complete approach to tackle infectious diseases, analysing and disseminating a wider range of information, which is necessary to properly support policy making in health care.

## Figures and Tables

**Figure 1 microorganisms-11-02496-f001:**
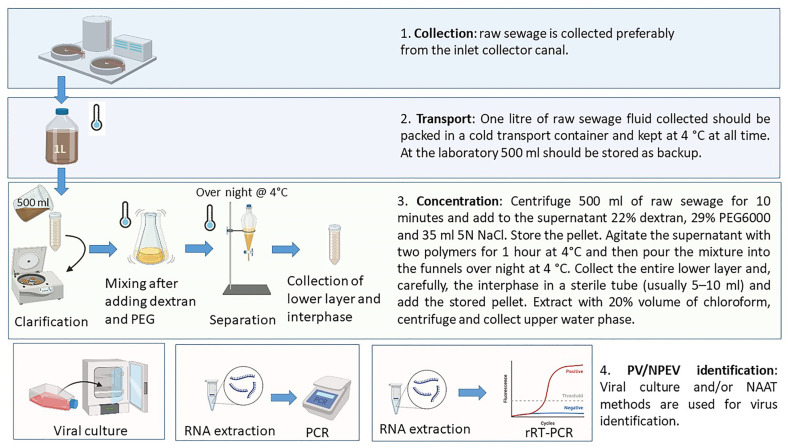
Schematic illustration of the process to concentrate and prepare raw sewage water samples for analytical assays. For more details on the procedure described in this figure, please refer to the Guidelines for Environmental Surveillance of Poliovirus Circulation [7] standard operational procedures (SOPs) for poliovirus detection and identification. Figure created with BioRender.com. Accessed on 25 September 2023.

**Table 1 microorganisms-11-02496-t001:** Brief description of the principal methods for sample preparation and viral detection described in this paper, reporting the overall timing and limitations.

	Method	Time	Limitations
Sample Preparation	Traditional concentration methods	2 days	Possible inhibition with molecular methods
	Ultracentrifugation	1–2 h	Possible lower concentration
Detection Methods	Cell culture methods	3–7 days	Selection of specific types; lack of availability of monoclonal antibodies for typing; type refractory for isolation cannot be detected; necessity of large volumes for inoculation; sample needs to contain viable virus to reach required sensitivity.
	Real-time PCR based methods	1–2 h	Possible inhibitions due to the sample matrix; miss detection if low concentration.
	Sanger-based methods	1–3 h	Possible missing of sequence of types that are less concentrated; needs primer and probe management for type identification; needs sequence editing and management to identify types.
	NGS-based methods	1–2 days based on the platform in use	Necessity of bioinformatics pipeline; higher costs; risk of generating low reads for types less concentrated in the sample.

## Data Availability

No new data were created or analysed in this study. Data sharing is not applicable to this article.

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
