# Peer review of "Wastewater Surveillance in Europe for Non-Polio Enteroviruses and Beyond"

_microorganisms, 2023, doi:10.3390/microorganisms11102496_

Round 1

Reviewer 1 Report

Summary and Recommendation

The authors describe the history of wastewater surveillance, its use for poliovirus, non-polio enteroviruses and SARS-CoV-2 surveillance. The authors have done a good job putting wastewater surveillance in perspective and clearly recommending future directions especially in the one-health framework as a tool for pandemic preparedness. The review deserves to be published with minor revision.

Minor Comments

1.     Please remove sectional headings like introduction, methods, results and discussion.

2.     There is mention of boxes 1-3 in text but only box 1 is present in the manuscript. Please correct this.

3.     Page 1, Line 28: Please consider changing ‘which are an increasingly recognised’ to ‘which are increasingly recognised’.

4.     Page 1, Line 37: Please consider changing ‘sequenced-based’ to ‘sequence-based’

5.     Page 4, Line 153: Please consider changing ‘to detect of branches of containment’ to ‘to detect any breach of containment’.

6.     Page 4, Line 164: Please consider changing ‘consists in a collection of’ to ‘consists in collection of’.

7.     Please cite the data presented in paragraph 2 of page 6.

8.     Page 7, Line 268: Please consider changing ‘The procedure is referred to the original guidelines’ to ‘For more details on the procedure described in this figure, please refer to the Guidelines’

9.     Page 7, Line 283: Please consider changing ‘to early detection’ to ‘early detection’.

10.  Page 7, Lines 288-295: Please merge both sentences into a paragraph.

11.  Page 7, Line 292: Please consider changing ‘wastewater surveillance of sewage water can’ to ‘wastewater surveillance can’.

12.  Page 8, Line 302: Please consider changing ‘may be also’ to ‘may also be’.

13.  Page 8, Line 303: Please consider changing ‘spillover in the’ to ‘spillover into the’.

Author Response

Dear reviewer, thank you very much for your suggestions. Please, consider the follow point-by-point reply here below and the modified version of the manuscript, where we highlighted the changes in blue (including those from other reviewers).

Best regards,

Laura on behalf of the authors

Minor Comments

  1. Please remove sectional headings like introduction, methods, results and discussion. Thank you for the suggestion we removed the headings.
  1. There is mention of boxes 1-3 in text but only box 1 is present in the manuscript. Please correct this.

The pdf shared with reviewer did not include the boxes for an error in the system out of our control. We hope you can review them in this updated version.

  1. Page 1, Line 28: Please consider changing ‘which are an increasingly recognised’ to ‘which are increasingly recognised’. Thank you for spotting this, we deleted it.
  2. Page 1, Line 37: Please consider changing ‘sequenced-based’ to ‘sequence-based’ Thank you, for spotting this, we modified it.
  3. Page 4, Line 153: Please consider changing ‘to detect of branches of containment’ to ‘to detect any breach of containment’. Thank you, we modified it accordingly.
  4. Page 4, Line 164: Please consider changing ‘consists in a collection of’ to ‘consists in collection of’. Thank you, for spotting this, we modified it.
  5. Please cite the data presented in paragraph 2 of page 6. Thank you, we added the reference to this sentence.
  6. Page 7, Line 268: Please consider changing ‘The procedure is referred to the original guidelines’ to ‘For more details on the procedure described in this figure, please refer to the Guidelines’ Thank you, we modified it accordingly.
  7. Page 7, Line 283: Please consider changing ‘to early detection’ to ‘early detection’. Thank you for spotting this, we deleted it.
  8. Page 7, Lines 288-295: Please merge both sentences into a paragraph. Thank you, we merged the two sentences.
  9. Page 7, Line 292: Please consider changing ‘wastewater surveillance of sewage water can’ to ‘wastewater surveillance can’. Thank you for spotting this, we deleted it.
  10. Page 8, Line 302: Please consider changing ‘may be also’ to ‘may also be’. Thank you, we modified it accordingly.
  11. Page 8, Line 303: Please consider changing ‘spillover in the’ to ‘spillover into the’. Thank you, we modified it accordingly.

Reviewer 2 Report

This report of an expert webinar on wastewater surveillance for enteroviruses and other pathogens suffers only from errors of formatting and referencing. This topic is well described and current methodologies and testing are discussed.

Although the authors discuss boxes 2 and 3 on lines 150-154 which should have examples of how surveillance can represent the whole population and branches of containment respectively, these boxes are not present in the manuscript.

Referencing:

Lines 56-57: “In fact, at least 30% of concentrated sewage from grab samples and at least 10% of traps should reveal non-polio enterovirus.” There should be a reference for this.

Lines 228-247: This discussion ends with “Results were published in a weekly report”. There should be a reference for this.

Reference 35 is not present in the text of the manuscript.

Typographic errors:

Line 211: “Few year ago” should be “A few years ago”

Line 365-369: There is an additional reference embedded in Reference 16.

Author Response

Dear reviewer, thank you very much for your suggestions. Please, consider the follow point-by-point reply here below and the modified version of the manuscript, where we highlighted the changes in blue (including those from other reviewers).

Best regards,

Laura on behalf of the authors

Although the authors discuss boxes 2 and 3 on lines 150-154 which should have examples of how surveillance can represent the whole population and branches of containment respectively, these boxes are not present in the manuscript. The pdf shared with reviewer did not include the boxes for an error in the system out of our control.

Referencing:

Lines 56-57: “In fact, at least 30% of concentrated sewage from grab samples and at least 10% of traps should reveal non-polio enterovirus.” There should be a reference for this. Thank you for the suggestion, we added the reference as requested

Lines 228-247: This discussion ends with “Results were published in a weekly report”. There should be a reference for this. Thank you, we added the reference as requested

Reference 35 is not present in the text of the manuscript. This reference was part of the boxes not shared in the pdf with reviewer. Apologise for the inconvenience.

Comments on the Quality of English Language

Typographic errors:

Line 211: “Few year ago” should be “A few years ago” Thank you, we corrected this sentence

Line 365-369: There is an additional reference embedded in Reference 16. Thank you, we corrected it.

Reviewer 3 Report

The manuscript entitledWastewater surveillance for non-polio enteroviruses and beyond” by Bubba and collaborators describes the wastewater-based surveillance system for PV detection in Europe, as well as its applications and limitations for surveillance of NPEVs. This manuscript is well-written and may be of broad interest to those looking to improve the wastewater surveillance of pathogens for public health. I have a few recommendations, which will be covered below:

1 – The authors discussed the wastewater surveillance system mainly in European countries. As a result, consider the possibility of including the word “Europe” in the title.

2- I couldn't find boxes 2 and 3.

3- Line 179: remove “hour” of the sentence.

4- Consider including a table comparing cell culture, RT-PCR, and NGS approaches for wastewater monitoring highlighting limitations, turnaround time, cost, and level of interpretation….

5- I believe that the authors can explore more thoroughly the manuscript. Thus,  consider including a brief description of the concerns associated with requirements of the standardized wastewater-monitoring techniques and the use of human fecal indicators, such as pepper mild mottle virus, in the context of NPEV surveillance.

6- Lines 246-247: consider including a reference to the statement: “Results were published in a weekly report”.

No comments

English ok

Author Response

Dear reviewer,

thank you very much for your suggestions. Please, consider the follow point-by-point reply here below and the modified version of the manuscript, where we highlighted the changes in blue (with other reviewers’ comments).

Best regards,

Laura on behalf of the authors

1 – The authors discussed the wastewater surveillance system mainly in European countries. As a result, consider the possibility of including the word “Europe” in the title. Thank you for the suggestion, we added it to the title

2- I couldn't find boxes 2 and 3. The pdf shared with reviewer did not include the boxes for an error in the system out of our control.

3- Line 179: remove “hour” of the sentence. Thank you for spotting this, we deleted it.

4- Consider including a table comparing cell culture, RT-PCR, and NGS approaches for wastewater monitoring highlighting limitations, turnaround time, cost, and level of interpretation…. Thank you for this suggestion. We prepared a table reporting both sample preparation methods and detection methods, as those can be combined differently based on the aim and capacity of the laboratory.

5- I believe that the authors can explore more thoroughly the manuscript. Thus,  consider including a brief description of the concerns associated with requirements of the standardized wastewater-monitoring techniques and the use of human fecal indicators, such as pepper mild mottle virus, in the context of NPEV surveillance. Thank you for this suggestion, we added two sentences at lines: 51-61 and 315-316

6- Lines 246-247: consider including a reference to the statement: “Results were published in a weekly report”. Thank you, we added the requested reference.
